# Differential Evolution with Shadowed and General Type-2 Fuzzy Systems for Dynamic Parameter Adaptation in Optimal Design of Fuzzy Controllers

**Patricia Ochoa, Oscar Castillo * , Patricia Melin and José Soria**

Tijuana Institute of Technology, Calzada Tecnologico s/n, Fracc. Tomas Aquino, 22379 Tijuana, Mexico;
martha.ochoa18@tectijuana.edu.mx (P.O.); pmelin@tectijuana.mx (P.M.); jsoria57@gmail (J.S.)
* Correspondence: ocastillo@tectijuana.mx

**Abstract:** This work is mainly focused on improving the differential evolution algorithm with the utilization of shadowed and general type 2 fuzzy systems to dynamically adapt one of the parameters of the evolutionary method. Previously, we have worked with both kinds of fuzzy systems in different types of benchmark problems and it has been found that the use of fuzzy logic in combination with the differential evolution algorithm gives good results. In some of the studies, it is clearly shown that, when compared to other algorithms, our methodology turns out to be statistically better. In this case, the mutation parameter is dynamically moved during the evolution process by using shadowed and general type-2 fuzzy systems. The main contribution of this work is the ability to determine, through experimentation in a benchmark control problem, which of the two kinds of the used fuzzy systems has better results when combined with the differential evolution algorithm. This is because there are no similar works to our proposal in which shadowed and general type 2 fuzzy systems are used and compared. Moreover, to validate the performance of both fuzzy systems, a noise level is used in the controller, which simulates the disturbances that may exist in the real world and is thus able to validate statistically if there are significant differences between shadowed and general type 2 fuzzy systems.

**Keywords:** shadowed type-2 fuzzy sets; generalized type-2 fuzzy systems; differential evolution algorithm

## 1. Introduction

The utilization of new strategies to improve the functioning of certain processes is something very common today, and under this concept we have the differential evolution (DE) algorithm, which is used in multiple disciplines to perform optimization. The main approach for this work is the adaptation of a parameter of the DE algorithm using two variants of fuzzy logic, which are shadowed and general type 2 fuzzy systems.

Previously, a study was carried out using the differential evolution algorithm and the concept of shadowed type 2 fuzzy systems applied to benchmark functions and a specific control problem [1]. In this work, we are now aiming at comparing the two concepts of shadowed and general type 2 fuzzy systems in order to find out which method is better for improving the performance of the DE algorithm in the process of optimizing fuzzy controllers.

Today, the utilization of shadowed type 2 fuzzy systems has become more common in the literature, and below we briefly mention some of these recent works in different disciplines. For example, a shadowed set-based method and its application to large-scale group decision making was proposed in [2], a more comprehensible perspective for interval shadowed sets obtained from fuzzy sets was put forward in [3], an interval data driven construction of shadowed sets with application to linguistic word modelling was outlined in [4], and a shadowed set approximation of fuzzy sets based on nearest quota of fuzziness

was described in [5]. In addition, an approach for parameterized shadowed type-2 fuzzy membership functions applied in control applications was outlined in [6], a two-threshold model for shadowed set with gradual representation of cardinality is presented in [7], a hybrid design of shadowed type-2 fuzzy inference systems applied in diagnosis problems was put forward in [8], shadowed sets as a way for representing and processing fuzzy sets is presented in [9], a sparse signal recovery for ultrasonic detection and reconstruction of shadowed flaws is shown in [10], a constrained three-way approximations of fuzzy sets: from the perspective of minimal distance in [11]. Moreover, a method for constructing shadowed sets and three-way approximations of fuzzy sets in [12], an entropy-based shadowed set approximation of intuitionistic fuzzy sets in [13], a game theoretic approach to shadowed sets is presented in [14], shadowed sets of dynamic fuzzy sets are presented in [15], a constrained shadowed sets and fast optimization algorithm are outlined in [16], fuzzy-entropy-based game theoretic shadowed sets are described in [17], and shadowed sets-based linguistic term modeling and its application in multi-attribute decision-making are studied in [18], just to mention some related papers.

In a similar fashion, the use of general type 2 fuzzy systems has become more common in different application areas, but mainly in the control area, and this work is mainly focused on this area. Some related works can be mentioned as follows: a dynamic general type-2 fuzzy system with optimized secondary membership for online frequency regulation is studied in [19], a hybridized forecasting method based on weight adjustment of neural network using generalized type-2 fuzzy set was outlined in [20], parameter adaptation in the imperialist competitive algorithm using generalized type-2 fuzzy logic was described in [21], the optimization of fuzzy controller design using a differential evolution algorithm with dynamic parameter adaptation based on type-1 and interval type-2 fuzzy systems was put forward in [22], a comprehensive review on type 2 fuzzy logic applications was outlined in [23], a dynamic general type-2 fuzzy system has been used with optimized secondary membership for online frequency regulation [24], while an intelligent oversampling approach based upon general type-2 fuzzy Sets was adopted to detect web spam [25]. A novel intuitionistic based interval type-2 fuzzy similarity measures with application to clustering in [26], a dynamic event-triggered sliding mode control for interval type-2 fuzzy systems with fading channels is shown in [27], a general type-2 fuzzy gain scheduling PID controller with application to power-line inspection robots in [28], an online general type-2 fuzzy classifier using evolving type-1 rules is shown in [29], input-to-state stabilization of interval type-2 fuzzy systems subject to cyberattacks with an observer-based adaptive sliding mode approach are studied in [30], general type-2 fuzzy logic systems based on shadowed sets are presented in [31], an adaptive type-2 fuzzy system is used for the synchronization and stabilization of chaotic non-linear fractional order systems in [32], and general interval approach for encoding words into interval type-2 fuzzy sets based on normal distribution and free parameter is adopted in [33].

In general, the most relevant contribution of the article is the comparison of the performance of shadowed type-2 and general type-2 fuzzy systems in achieving dynamic parameter adaptation in DE for improving its performance. This was achieved by making a comparison regarding the performance of the DE in optimizing a fuzzy controller applied to a nonlinear plant. A statistical comparison was used to verify which of the two types of fuzzy system is better for dynamic parameter adjustment in the DE algorithm. It can be mentioned that this has not been previously done in the current literature, so in this sense it is a novel work.

The article contains the following sections: Section 2 summarizes the basic constructs of shadowed type-2 fuzzy systems theory, Section 3 outlines the general type-2 fuzzy systems theory, Section 4 explains in detail the differential evolution algorithm, Section 5 explains the method for dynamic parameter adjustment in differential evolution, and Section 6 shows the experimentation performed with the control problem. In Section 7, a discussion of results is presented, and in Section 8 the conclusions are offered, as well as some possible lines of future research work.

## 2. Type-2 Fuzzy Systems and Shadowed Sets

In the literature, the fuzzy set term appears for the first time in 1965 [34], which mainly tells us that, as system complexity increases, the preciseness of its perception and our ability to express its behavior decreases, and it is from this idea that fuzzy systems emerge. However, today the fuzzy systems that we are now dealing with have evolved to what we know as general type-2 fuzzy systems, which can help to solve more complex systems or with higher uncertainty. The mathematical formulation of general type-2 fuzzy sets is expressed in Equation (1):

$$\widetilde{\widetilde{A}} = \left\{ \left( (x,u), u_{\widetilde{\widetilde{A}}}(x) \right) \big| \forall x \in X, \ \forall u \in J_x^u \subseteq [0,1] \right\} \tag{1}$$

The general type-2 fuzzy set (GT2 FS) is currently used in different real-world applications, and there are some options to model or approximate a GT2 FS, and one of them can be the vertical slices or z-slices representation [35–37]. The main part of this work focuses on the continuation of the previous work on fuzzy systems for dynamic parameter adaptation in harmony search and differential evolution. We continue taking into account the main idea of the previous work, which focuses on the representation of $\alpha$ planes, which mainly tells us that we can discretize the secondary axis of GT2 FS in several horizontal sections, which are called $\alpha$ planes. These $\alpha$ planes are expressed by Equation (2) and can be calculated as an interval type-2 fuzzy system (IT2FIS) [38]. Equation (3) expresses the modeling of a general fuzzy inference system (GT2FIS) as the union of the IT2FIS.

$$\widetilde{A}_\alpha = \{((x,u), \alpha) | \forall x \in X, \ \forall u \in J_x \subseteq [0,1]\} \tag{2}$$

$$\widetilde{\widetilde{A}} = \cup \widetilde{A}_\alpha \tag{3}$$

The purpose of the shadowed type-2 FIS [39] consists on reducing the computational cost represented by the use of $\alpha$-planes, and the main idea of this proposal is to model the GT2FIS with only two optimal $\alpha$-planes, eliminating the excessive precision. The aforementioned knowledge is based on the concepts proposed by Pedrycz, who proposed the theory of shadowed sets in [40–42].

Equation (4) expresses in detail the shadowed set concept, which consists of performing two $\alpha$-cuts on a fuzzy set, with $\alpha$ and $\beta$ values which are based on these $\alpha$-cuts.

$$S_{\mu_A}(x) = \begin{cases} 1, \ if \ \mu_A(x) \geq \alpha \\ 0, \ if \ \mu_A(x) \leq \beta \\ [0,1], \ if \ \alpha \leq \mu_A(x) \geq \beta \end{cases} \tag{4}$$

There are 3 regions, which can present the following interpretations:
- The elevated region for the membership degrees with a value of 1.
- The reduced region for the membership degrees with a value of 0.
- The shaded region with degree of membership in [0, 1].

Using these regions as a reference, Pedrycz proposes that for finding the optimal $\alpha$ and $\beta$ values, they can be calculated using Equation (5), which expresses the calculation to obtain the shadowed area.

$$elevated \ area_{(\alpha,\beta)}(\mu_A) + reduced \ area_{(\alpha,\beta)}(\mu_A) = shadowed \ area_{(\alpha,\beta)}(\mu_A) \tag{5}$$

The aforementioned is represented graphically with Figure 1.

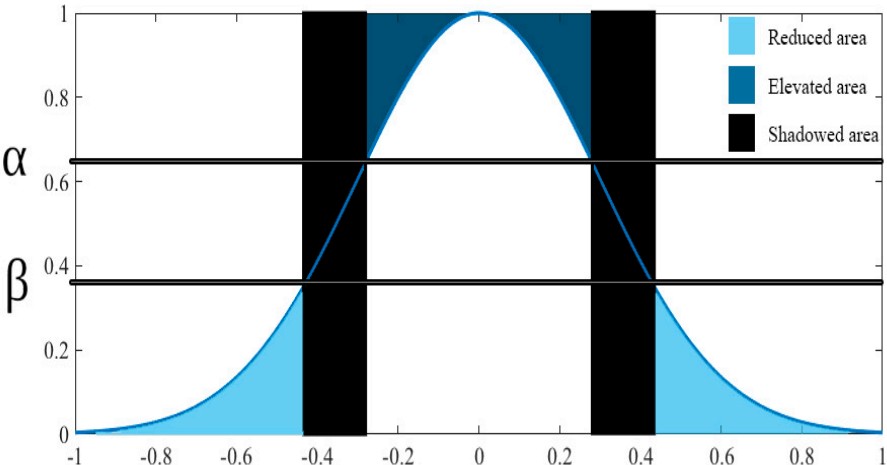

**Figure 1.** Graphical representation of a shadowed set.

The optimal $\alpha$ and $\beta$ values can then be obtained by optimizing the $V(\alpha, \beta)$ function described by Equation (6)

$$V(\alpha, \beta) = \left| \int_{x \in A_r} \mu_A(x)dx + \int_{x \in A_e} (1 - \mu_A(x))dx - \int_{x \in S} dx \right| \tag{6}$$

This is how we can take advantage of shadowed type-2 fuzzy sets to combine them into the structure of the differential evolution algorithm for dynamic parameter adaptation.

A fuzzy system can be built with a trapezoidal shadowed type-2 fuzzy set membership function (TrapG ST2 MF) introduced in [43] and that is based on a trapezoidal general type-2 (GT2) membership function with a Gaussian membership function as a secondary membership function. The mathematical knowledge of the membership functions is formulated in Equation (7) and we can appreciate its graphical form in Figure 2.

$$\text{TrapG ST2 MF} = \begin{cases} \propto_o \begin{cases} \overline{\mu}_O = \frac{\overline{\mu}_t(x) + \underline{\mu}_t(x)}{2} - 1.449 \left| \frac{\overline{\mu}_t(x) - \underline{\mu}_t(x)}{10} \right| \\ \underline{\mu}_O = \frac{\overline{\mu}_t(x) + \underline{\mu}_t(x)}{2} + 1.449 \left| \frac{\overline{\mu}_t(x) - \underline{\mu}_t(x)}{10} \right| \end{cases} \\ \propto_l \begin{cases} \overline{\mu}_I = \frac{\overline{\mu}_t(x) + \underline{\mu}_t(x)}{2} - 0.9282 \left| \frac{\overline{\mu}_t(x) - \underline{\mu}_t(x)}{10} \right| \\ \underline{\mu}_I = \frac{\overline{\mu}_t(x) + \underline{\mu}_t(x)}{2} + 0.9282 \left| \frac{\overline{\mu}_t(x) - \underline{\mu}_t(x)}{10} \right| \end{cases} \end{cases} \tag{7}$$

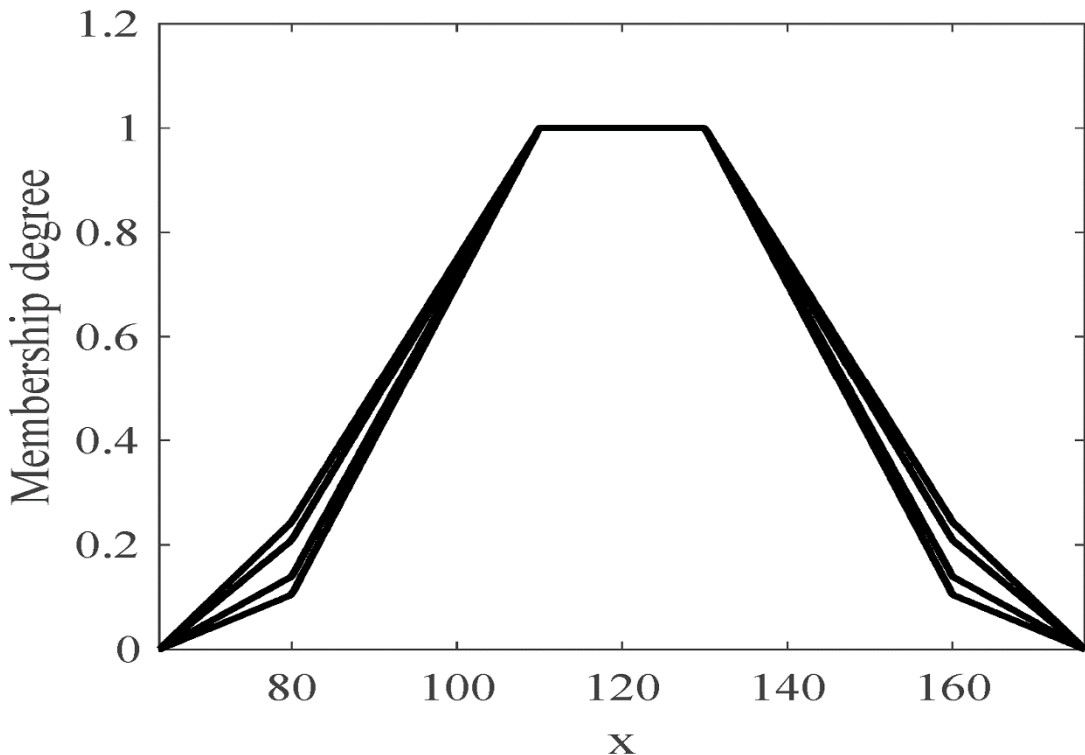

**Figure 2.** Graphical representation of a Trapezoidal ST2 MF.

### 3. General Type-2 Fuzzy Systems

Another important part of our work is the utilization of general type-2 fuzzy logic, which works under the same concept as Type-1 and interval type-2 fuzzy logic systems, except that their mathematical functions contemplate different concepts since GT2FSs are well known for handling higher levels of uncertainty. There are different definitions about the mathematical functions used in a general type-2 fuzzy logic system, and for this work we are going to use the notation presented on [44–47]. The formulation of general type-2 fuzzy sets is presented in Equation (8).

$$\widetilde{\widetilde{A}} = \left\{ \left( (x,u), \mu_{\widetilde{A}}\left(x,u\right) \right) \mid \forall_x \in X, \ \forall_u \in J_x \subseteq [0,1] \right\} \tag{8}$$

where $J_x \subseteq [0, 1]$, $x$ represents a primary membership function partition, and $u$ represents a secondary membership function partition.

The graphical representation of a type-2 membership function is illustrated in Figure 3. On the other hand, we can notice the concept of footprint uncertainty (FOU) in Figure 4, which is shown in the third dimension and enables a clearer visualization of the real-world uncertainty modeling.

There is a difference in the nomenclature of each of the fuzzy systems:

The notation $\mu(x)$ is used for type-1 and interval type-2 fuzzy systems.

The notation $f_x(u)$ is used for general type-2 fuzzy logic systems.

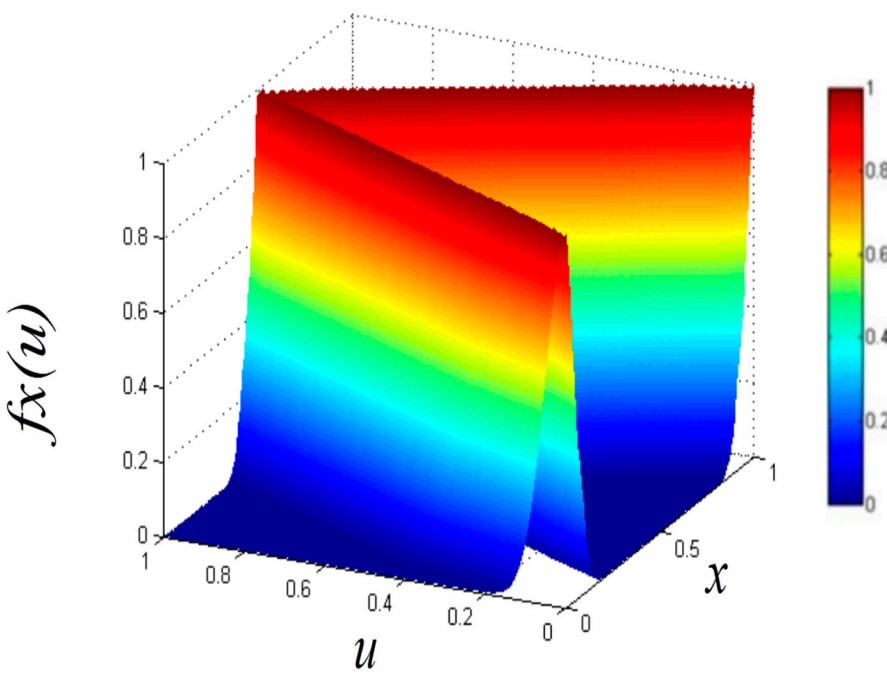

**Figure 3.** Visual representation of the GT2FS membership function.

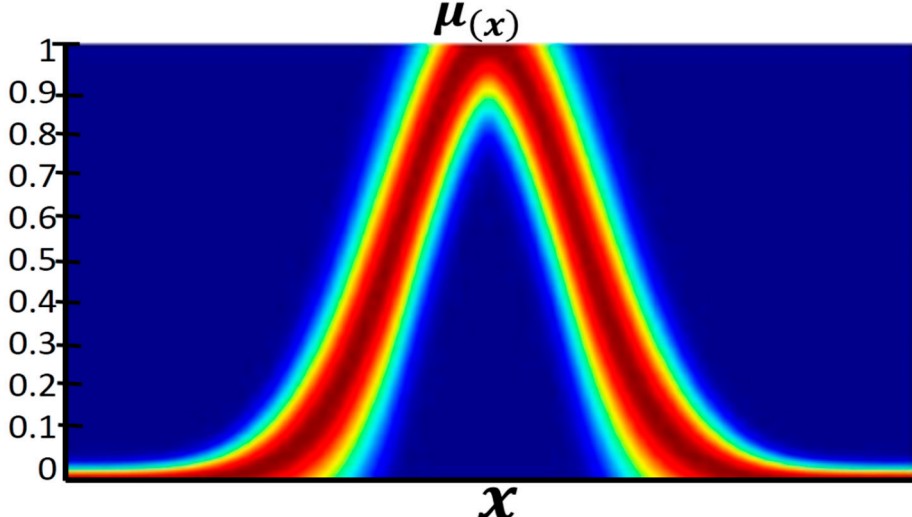

**Figure 4.** Visual representation of the FOU of the GT2FS membership function.

The $\alpha$-plane for a general type-2 fuzzy set, in this case $\widetilde{A}$, is denoted by $\tilde{A}_{\alpha}$, and it is the union of all primary membership functions of $\tilde{A}$, which secondary membership degrees are higher or equal to $\alpha$ ($0 \leq \alpha \leq 1$) [48,49]. The visual representation of an alpha plane can be found in Figure 5, in the same way the expression of the alpha plane is given by Equation (9).

$$\widetilde{A}_{\alpha} = \left\{ (x, u), \mu_{\widetilde{A}}(x, u) \geq \alpha \middle| \forall_x \in X, \forall_u \in J_x \subseteq [0, 1] \right\} \tag{9}$$

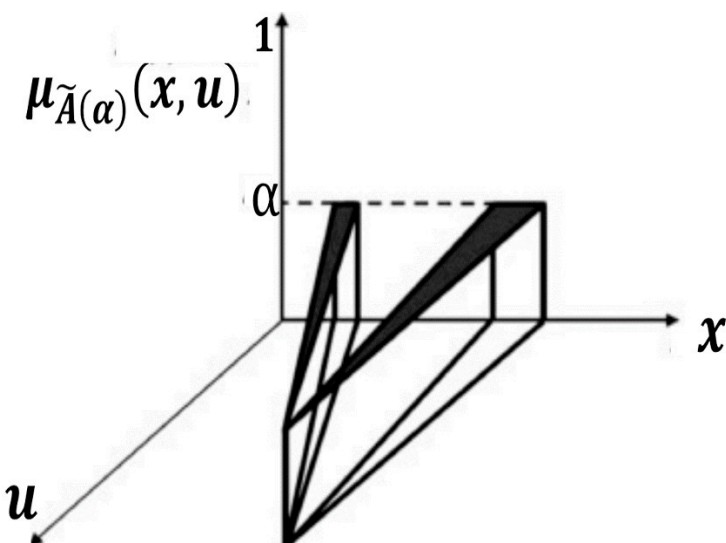

**Figure 5.** Representation of an alpha-plane corresponding to a type-2 fuzzy set.

## 4. Differential Evolution Algorithm

Differential evolution is a metaheuristic with which we have previously worked, and it has always provided good results in the different experiments that we have performed for different applications. This is an algorithm that is mainly composed of the following operations:

Equations (10)–(15) define the initialization of the population structure, Equation (16) represents the initialization of the algorithm, Equation (17) represents the mutation performed by the algorithm, Equation (18) shows the crossover process, and finally Equation (19) expresses the last step, which is the selection operator.

A more detailed explanation of the equations can be found in previous works [50–54].

**Structure of the Population**

$$P_{x,g} = (x_{i,g}), \; i = 0, 1, \ldots, Np - 1, \; g = 0, 1, \ldots, g_{max}, \tag{10}$$

$$x_{i,g} = (x_{j,i,g}), \; j = 0, 1, \ldots, D - 1 \tag{11}$$

$$P_{v,g} = (v_{i,g}), \; i = 0, 1, \ldots, Np - 1, \; g = 0, 1, \ldots, g_{max}, \tag{12}$$

$$v_{i,g} = (v_{j,i,g}), \; j = 0, 1, \ldots, D - 1 \tag{13}$$

$$P_{u,g} = (u_{i,g}), \; i = 0, 1, \ldots, Np - 1, \; g = 0, 1, \ldots, g_{max}, \tag{14}$$

$$u_{i,g} = (u_{j,i,g}), \; j = 0, 1, \ldots, D - 1 \tag{15}$$

**Initialization**

$$x_{j,i,0} = rand_j(0, 1) \cdot (b_{j,U} - b_{j,L}) + b_{j,L} \tag{16}$$

**Mutation**

$$v_{i,g} = x_{r_0,g} + F \cdot (x_{r_1,g} - x_{r_2,g}) \tag{17}$$

**Crossover**

$$u_{i,g} = u_{j,i,g} \begin{cases} v_{j,i,g} \; if \; (rand_j(0,1) \leq Cr \; or \; j = j_{rand}) \\ x_{j,i,g} \quad otherwise \end{cases} \tag{18}$$

**Selection**

$$x_{i,g+1} = \begin{cases} u_{i,g} \; if \; f(u_{i,g}) \leq f(x_{i,g}) \\ x_{i,g} \; otherwise \end{cases} \tag{19}$$

## 5. Differential Evolution Algorithm with Dynamic Parameter Adaptation

The structure of each of the fuzzy systems created for experimentation is explained in more detail below. We consider shadowed and general type-2 fuzzy systems, which contain one input and one output. As input variable we consider the "generations", which is represented in Equation (20), the experiment refers to generations for the Fuzzy DE. In this case, the current experiment represents the current generation number, and the maximum of experiments represents the maximum number of generations. For the output parameter, we are using the variable F representing the mutation parameter of the differential evolution algorithm.

$$Generations = \frac{Current\ generation}{Maximun\ of\ generation} \tag{20}$$

Equation (21) represents the mutation parameter, and this parameter is the output of the fuzzy system. In other words, *F* is the fuzzy parameter, which changes dynamically in the DE.

$$F = \frac{\sum_{i=1}^{r_F} \mu_i^F\ (F_{1i})}{\sum_{i=1}^{r_F} \mu_i^F} \tag{21}$$

where *F* is the output and the mutation parameter; $r_{hmr}$ is the number of rules of the fuzzy systems corresponding to *F*; $F_{1i}$ is the output result for rule *i*; $\mu_i^F$ is the membership function of rule *i*.

The inputs and outputs of both fuzzy systems are granulated into three membership functions, and they are called low, medium, and high.

The rules that make both systems are based on previous experimentation experience, and these rules can be observed in Table 1.

**Table 1.** Rules of the ST2FDE fuzzy system.

| Generation | F | | |
|:---:|:---:|:---:|:---:|
| | **Low** | **Medium** | **High** |
| **Low** | – | – | Low |
| **Medium** | – | Medium | – |
| **High** | High | – | – |

➢ **Shadowed Type 2 fuzzy systems**

At first instance, we have a fuzzy system, which we called ST2FDE since it represents the fuzzy system using shadowed type-2 fuzzy sets. This fuzzy system is composed of an input called generations and an output called F that represents mutation in the differential evolution algorithm. Another characteristic of the system is that it corresponds to a Mamdani type, the Figure 6 shows the structure of the Shadowed Type-2 fuzzy system.

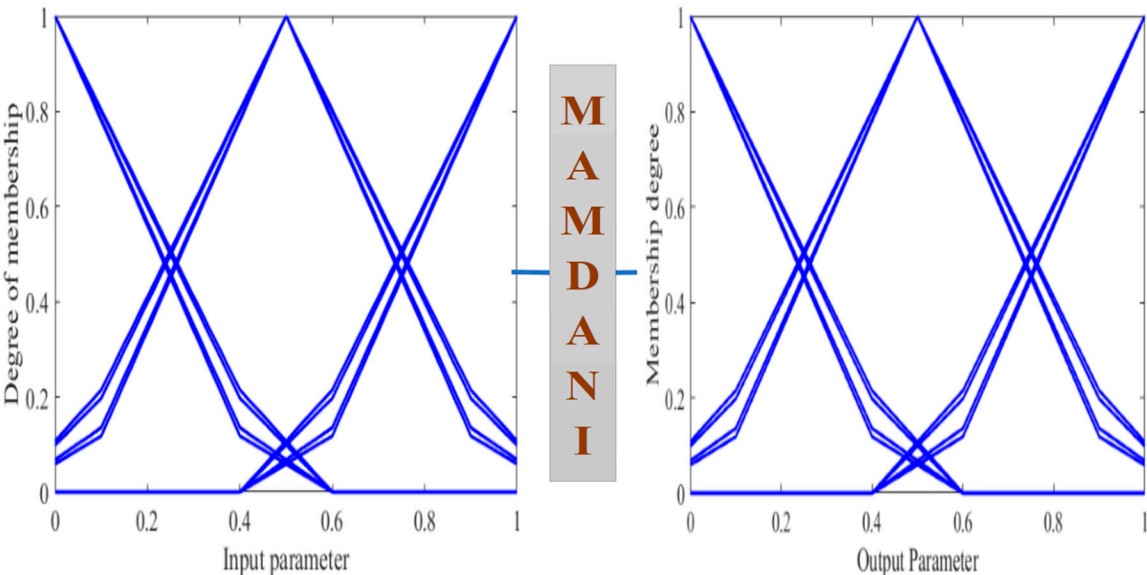

**Figure 6.** Structure of the Shadowed Type-2 fuzzy system.

➢ **General Type 2 fuzzy systems**

The second fuzzy system used in this paper is of general type-2 fuzzy form, just like our ST2FDE system contains an input called generations and an output called F that corresponds to the mutation. The type of membership functions that the system contains are triangular and their mathematical knowledge is expressed in Equation (22). We denote the general type-2 fuzzy system as GT2FDE.

$$
\begin{aligned}
\mu(x,u) &= trigausstype2(x,u[a_1,\ b_1,c_1,\ a_2,\ b_2,c_2,\rho]) \\
\mu(x,u) &= exp\left[-\tfrac{1}{2}\left(\tfrac{u-P_X}{\sigma_u}\right)\right] where \\
\mu_1(x) &= \max\left(\min\left(\tfrac{x-a_1}{b_1-a_1},\tfrac{c_1-x}{c_1-b_1}\right),0\right) and \\
\mu_2(x) &= \max\left(\min\left(\tfrac{x-a_2}{b_2-a_2},\tfrac{c_2-x}{c_2-b_2}\right),0\right) \\
\overline{\mu}(x) &= \begin{cases} \max(\mu_1(x),\ \mu_2(x))\ \forall_x\ \notin (b_1,b_2) \\ 1 \qquad\qquad\qquad\qquad \forall_x\ \in (b_1,b_2) \end{cases} \\
\underline{\mu}(x) &= \min(\mu_1(x),\ \mu_2(x)) \\
\rho_x &= \max\left(\min\left(\tfrac{x-a_x}{b_x-a_x}\right),\left(\tfrac{c_x-x}{c_x-b_x}\right),\ 0\right),\ where \\
a_x &= \tfrac{a_1+a_2}{2},\ b_x = \tfrac{b_1+b_2}{2},\ c_x = \tfrac{c_1+c_2}{2}, \\
\delta &= \overline{\mu}(x) - \underline{\mu}(x) \\
\sigma_u &= \tfrac{1+\rho}{2\sqrt{3}}\ \delta + \varepsilon
\end{aligned}
\tag{22}
$$

where $a_1$, $b_1$, and $c_1$ are the upper membership function parameters and $a_2$, $b_2$, *and* $c_2$ are the lower membership function parameters, respectively. In addition, $\rho$ is the fraction of uncertainty of the secondary membership function support, the Figure 7 shows the Structure of the General Type-2 fuzzy logic system.

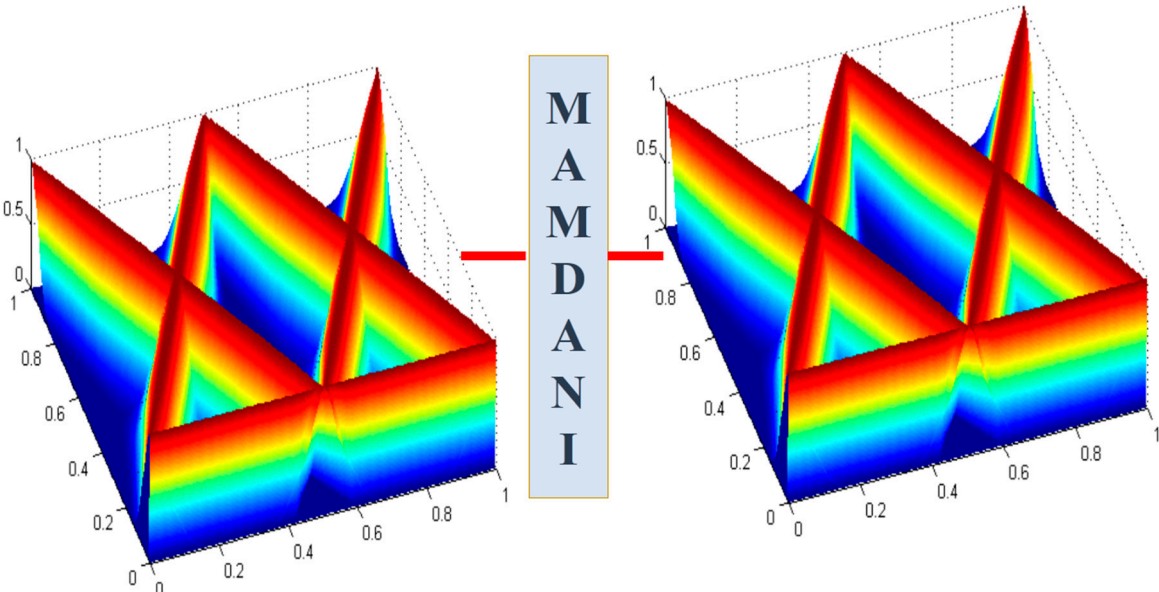

**Figure 7.** Structure of the General Type-2 fuzzy logic system.

Table 2 shows the mathematical expression used by the GT2FDE fuzzy system. This table summarizes the parameterized knowledge of the fuzzy system.

**Table 2.** Parameters of the General Type-2 fuzzy sets.

| | **Generalized Type-2 Fuzzy Logic Sets** |
|---|---|
| **Low** | $\mu_1(x) = \max\left(\min\left(\frac{x-0.5}{-0.08+0.5}, \frac{0.4-x}{0.4+0.08}\right), 0\right)$ *and* <br> $\mu_2(x) = \max\left(\min\left(\frac{x+0.4}{0.08+0.4}, \frac{0.5-x}{0.5-0.08}\right), 0\right)$ <br> $\overline{\mu}(x) = \begin{cases} \max(\mu_1(x),\ \mu_2(x))\ \forall_x\ \notin(-0.08,\ 0.08) \\ 1 \qquad\qquad\qquad \forall_x\ \in(-0.08,\ 0.08) \end{cases}$ <br> $\underline{\mu}(x) = \min(\mu_1(x),\ \mu_2(x))$ <br> $\rho_x = \max\left(\min\left(\frac{x-a_x}{b_x-a_x}\right), \left(\frac{c_x-x}{c_x-b_x}\right),\ 0\right),\ where$ <br> $a_x = \frac{-0.5-0.4}{2},\ b_x = \frac{-0.8-0.08}{2},\ c_x = \frac{-0.4-0.5}{2},$ <br> $\delta = \overline{\mu}(x) - \underline{\mu}(x)$ <br> $\sigma_u = \frac{1+\rho}{2\sqrt{3}}\delta + \varepsilon$ <br> *Where* $\rho = 0.5$ |
| **Medium** | $\mu_1(x) = \max\left(\min\left(\frac{x+0.084}{0.4+0.084}, \frac{0.92-x}{0.92-0.4}\right), 0\right)$ *and* <br> $\mu_2(x) = \max\left(\min\left(\frac{x-0.084}{0.5-0.084}, \frac{1.07-x}{1.07-0.5}\right), 0\right)$ <br> $\overline{\mu}(x) = \begin{cases} \max(\mu_1(x),\ \mu_2(x))\ \forall_x\ \notin(0.4,\ 0.5) \\ 1 \qquad\qquad\qquad \forall_x\ \in(0.4,\ 0.5) \end{cases}$ <br> $\underline{\mu}(x) = \min(\mu_1(x),\ \mu_2(x))$ <br> $\rho_x = \max\left(\min\left(\frac{x-a_x}{b_x-a_x}\right), \left(\frac{c_x-x}{c_x-b_x}\right),\ 0\right),\ where$ <br> $a_x = \frac{-0.084+0.084}{2},\ b_x = \frac{0.4-0.5}{2},\ c_x = \frac{0.92-1.09}{2},$ <br> $\delta = \overline{\mu}(x) - \underline{\mu}(x)$ <br> $\sigma_u = \frac{1+\rho}{2\sqrt{3}}\delta + \varepsilon$ <br> *Where* $\rho = 0.5$ |

**Table 2.** *Cont.*

| Generalized Type-2 Fuzzy Logic Sets | |
|---|---|
| **High** | $\mu_1(x) = \max\left(\min\left(\frac{x-0.4}{0.92-0.4}, \frac{1.4-x}{1.4-0.92}\right), 0\right)$ *and* <br> $\mu_2(x) = \max\left(\min\left(\frac{x-0.5}{1.07-0.5}, \frac{1.5-x}{1.5-1.07}\right), 0\right)$ <br> $\overline{\mu}(x) = \begin{cases} \max(\mu_1(x), \mu_2(x)) & \forall_x \notin (0.92, 1.07) \\ 1 & \forall_x \in (0.92, 1.07) \end{cases}$ <br> $\underline{\mu}(x) = \min(\mu_1(x), \mu_2(x))$ <br> $\rho_x = \max\left(\min\left(\frac{x-a_x}{b_x-a_x}\right), \left(\frac{c_x-x}{c_x-b_x}\right), 0\right), where$ <br> $a_x = \frac{0.4+0.5}{2}, \ b_x = \frac{0.92-1.07}{2}, \ c_x = \frac{1.4-1.5}{2},$ <br> $\delta = \overline{\mu}(x) - \underline{\mu}(x)$ <br> $\sigma_u = \frac{1+\rho}{2\sqrt{3}}\delta + \varepsilon$ <br> *Where* $\rho = 0.5$ |

## 6. Experiments Whit the D.C. Motor Speed Controller

For the experimentation, it was decided to use a benchmark control problem, which is used in real applications in industry. We decided to use the direct current (D.C.) motor speed control problem [55], and the purpose of the experimentation is to improve the response capacity of the motor by using the two proposals for fuzzy systems, namely GT2FDE and ST2FDE.

The structure of the controller with respect to the fuzzy system is of two inputs, which are the error and the error change, and the output that corresponds to the voltage. The controller is of the Mamdani type, and the aforementioned description can be appreciated in Figure 8. Another important aspect of the controller is its rules, which are represented in Table 3, and they form a rule base of 15 fuzzy rules.

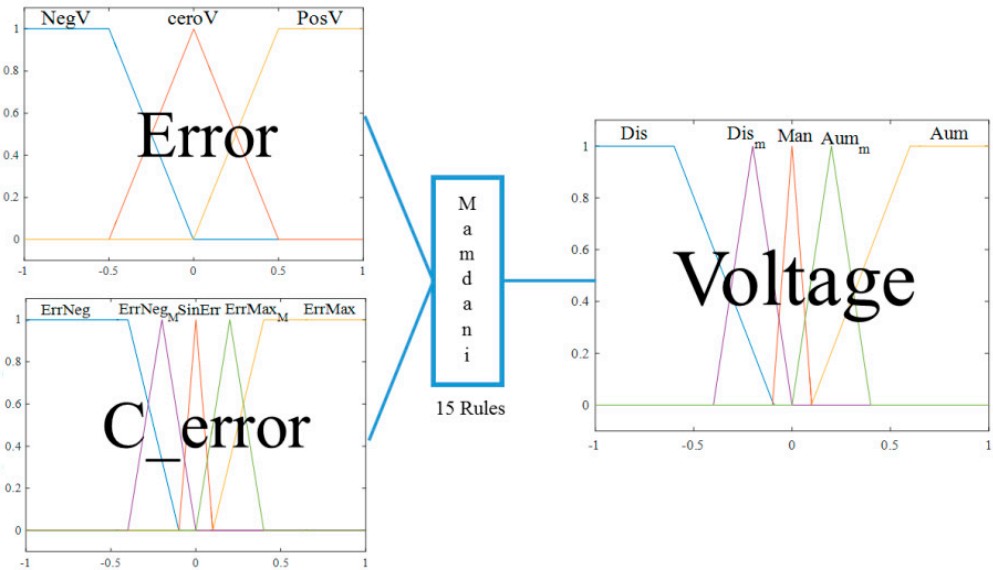

**Figure 8.** The structure of the fuzzy controller of the Motor.

**Table 3.** Fuzzy Rules for Motor Control.

| No. | Inputs | | Output |
| --- | --- | --- | --- |
| | **Error** | **Change in Error** | **Voltage** |
| 1 | NegV | ErrNeg | Dis |
| 2 | NegV | SinErr | Dis |
| 3 | NegV | ErrMax | Dis_m |
| 4 | ZeroV | ErrNeg | Aum_m |
| 5 | ZeroV | ErrMax | Dis_m |
| 6 | PosV | ErrNeg | Aum_m |
| 7 | PosV | SinErr | Aum |
| 8 | PosV | ErrMax | Aum |
| 9 | ZeroV | SinErr | Man |
| 10 | NegV | ErrNeg_M | Dis |
| 11 | ZeroV | ErrNeg_M | Aum_m |
| 12 | PosV | ErrNeg_M | Aum |
| 13 | PosV | ErrMax_M | Aum |
| 14 | ZeroV | ErrMax_M | Dis_m |
| 15 | NegV | ErrMax_M | Dis |

The main characteristic of the controller is to achieve moving from a resting state to a desired reference, which is a speed of 40 m/s. Figure 9 illustrates the reference for the speed of the controller with respect to time.

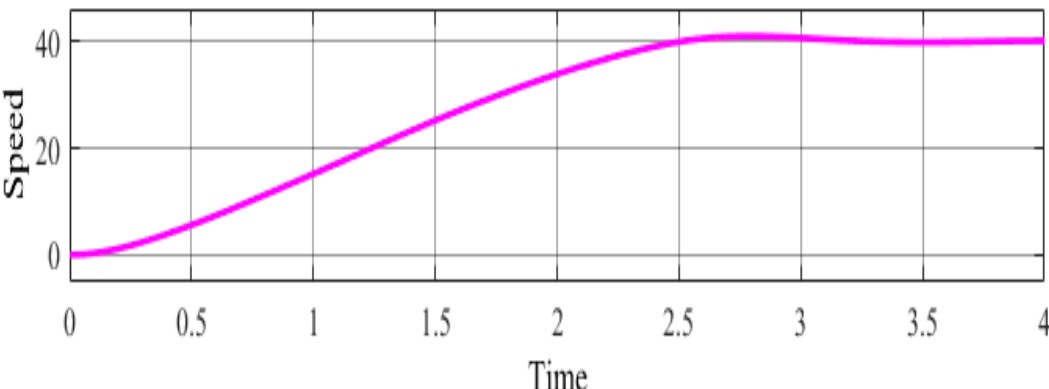

**Figure 9.** Speed response without optimization.

The experimentation of this work is mainly based on separately using the two fuzzy systems to optimize membership function parameters of the fuzzy system of the controller. The fuzzy controller is formed by 45 parameters that represent the sum of the points that make up each of the membership functions.

Figure 10 expresses the composition of the complete vector formed by all the fuzzy system parameters, and based on these parameters the evolutionary algorithm combined with the fuzzy system searches for the best architecture for the fuzzy controller.

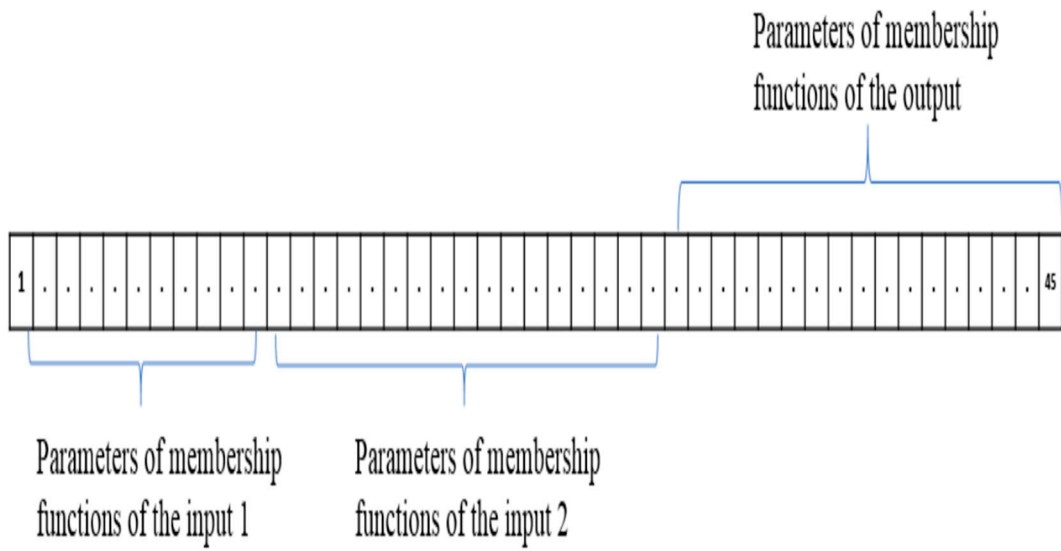

**Figure 10.** Chromosome for the fuzzy controller (membership functions parameters).

Figure 11 illustrates the workflow with which the experimentation was carried out. The differential evolution algorithm is initialized, which uses a fuzzy system to optimize the parameters of the controller's membership functions, and this process is repeated until the criterion of stop established in the DE.

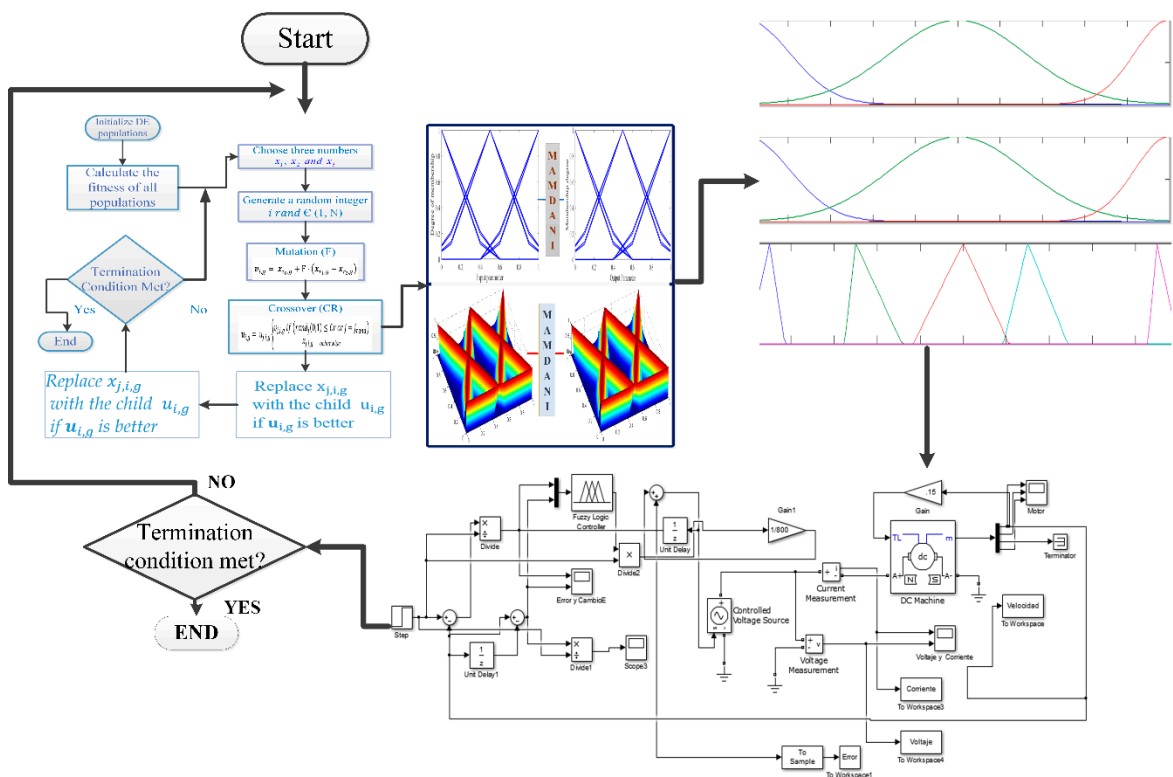

**Figure 11.** Structure of the experimentation process.

The experimentation was performed using the parameters shown in Table 4, and to validate which of the two proposed fuzzy systems has better performance, we decided to add a noise level to the controller. The different noise levels applied to this controller are: 0.5, 0.7, and 0.9 (Gaussian random number).

**Table 4.** Parameters of the algorithm.

| Parameters | ST2FDE and GT2FDE |
|---|---|
| Population | 50 |
| Dimensions | 45 |
| Generations | 30 |
| Number of experiments | 30 |
| F | Dynamic |
| Cr | 0.3 |

In this case, the objective function is defined by the root mean square error (RMSE) of the real values with respect to the reference speed for the motor, as illustrated in Equation (23):

$$RMSE = \sqrt{\frac{1}{N} \sum_{t=1}^{N} (x_t - \hat{x}_t)^2} \tag{23}$$

The 30 experiments were carried out applying each of the fuzzy systems varying the level of noise (0.5, 0.7, and 0.9). From these, the best results, the worst results, averages, and standard deviations were obtained.

Table 5 summarizes the aforementioned information from the experimentation using the shadowed type-2 fuzzy system (ST2FDE).

**Table 5.** Comparison of results using ST2FDE optimization of the fuzzy controller.

| | ST2FDE | | | |
|---|---|---|---|---|
| **Method** | **ST2FDE without Noise FLC** | **ST2FDE with Noise 0.5 FLC** | **ST2FDE with Noise 0.7 FLC** | **ST2FDE with Noise 0.9 FLC** |
| **Best** | $9.66\times10^{-01}$ | $9.41\times10^{-01}$ | $5.59\times10^{-01}$ | $4.52\times10^{-01}$ |
| **Worst** | $9.98\times10^{-01}$ | $9.96\times10^{-01}$ | $6.11\times10^{-01}$ | $6.56\times10^{-01}$ |
| **Average** | $9.84\times10^{-01}$ | $9.73\times10^{-01}$ | $5.86\times10^{-01}$ | $5.81\times10^{-01}$ |
| **Std.** | $8.45\times10^{-03}$ | $1.17\times10^{-02}$ | $1.40\times10^{-02}$ | $6.13\times10^{-02}$ |

The visual representation of the best results obtained by the performed experimentation with the fuzzy ST2FDE system is presented in Figures 12–15. These figures show the controller simulation with the different variants that we used. In these figures, the *x*-axis is the time measured in seconds and the *y*-axis is the speed measured in radians per second.

Figure 12 represents the simulation of the best result, $9.66\times10^{-01}$. This value can be seen in Table 5 and the simulation represents the simulation without noise in the controller using ST2FDE.

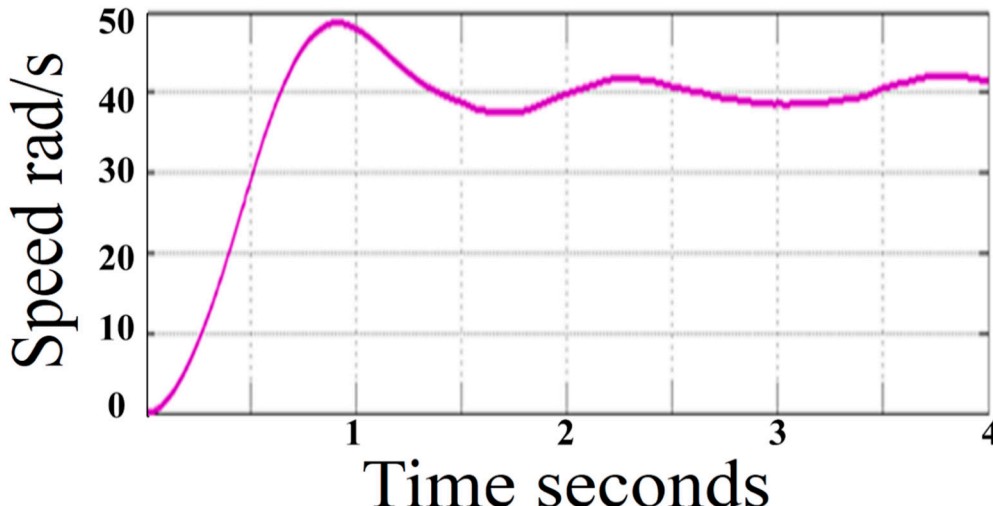

**Figure 12.** Simulation of the best result for ST2FDE without noise FLC.

Figure 13 represents the simulation of the best result, $9.41 \times 10^{-01}$ This value can be seen in Table 5 and the simulation represents the simulation with noise of 0.5 in the controller using ST2FDE.

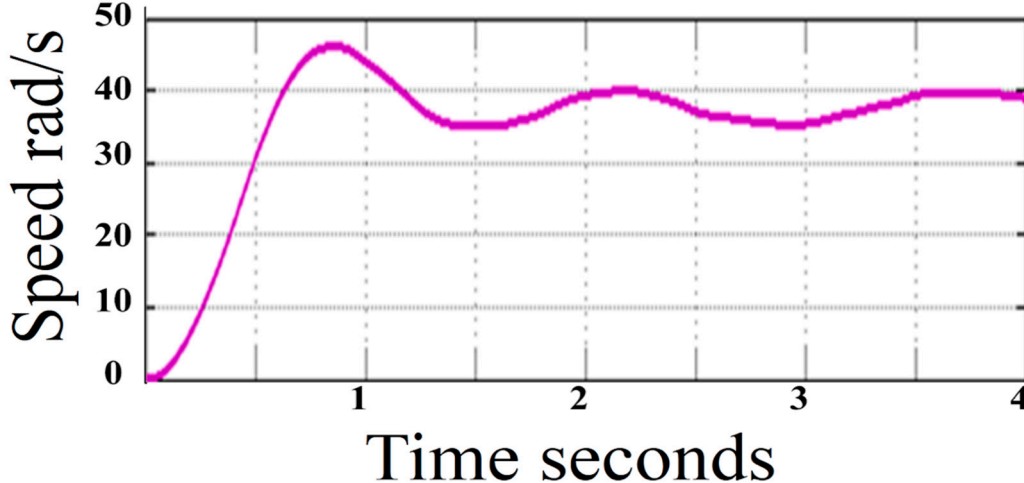

**Figure 13.** Simulation of the best result for ST2FDE with noise 0.5 FLC.

Figure 14 represents the simulation of the best result, $5.59 \times 10^{-01}$ This value can be seen in Table 5, and the simulation represents the simulation with noise of 0.7 in the controller using ST2FDE.

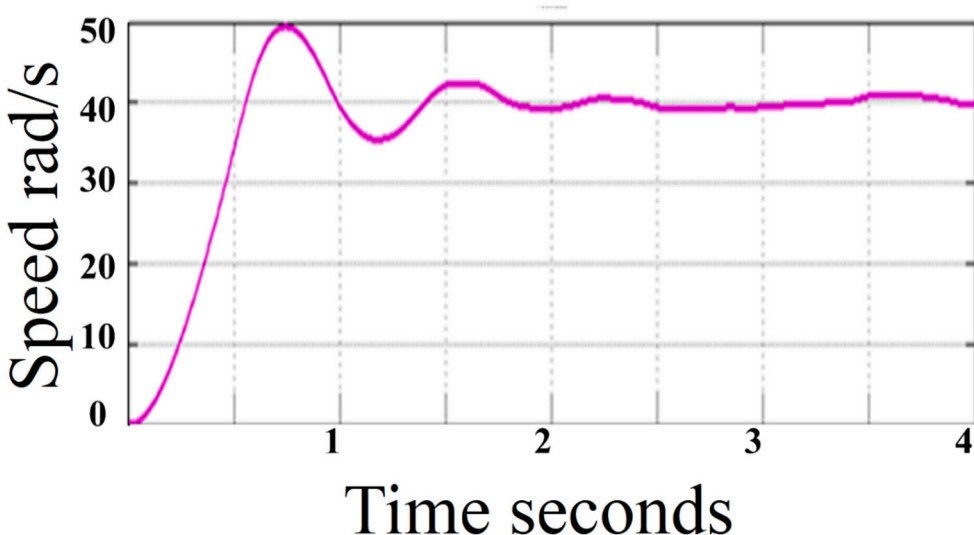

**Figure 14.** Simulation of the best result for ST2FDE with noise 0.7 FLC.

Figure 15 represents the simulation of the best result, $4.52 \times 10^{-01}$ This value can be seen in Table 5, and the simulation represents the simulation with noise of 0.9 in the controller using ST2FDE.

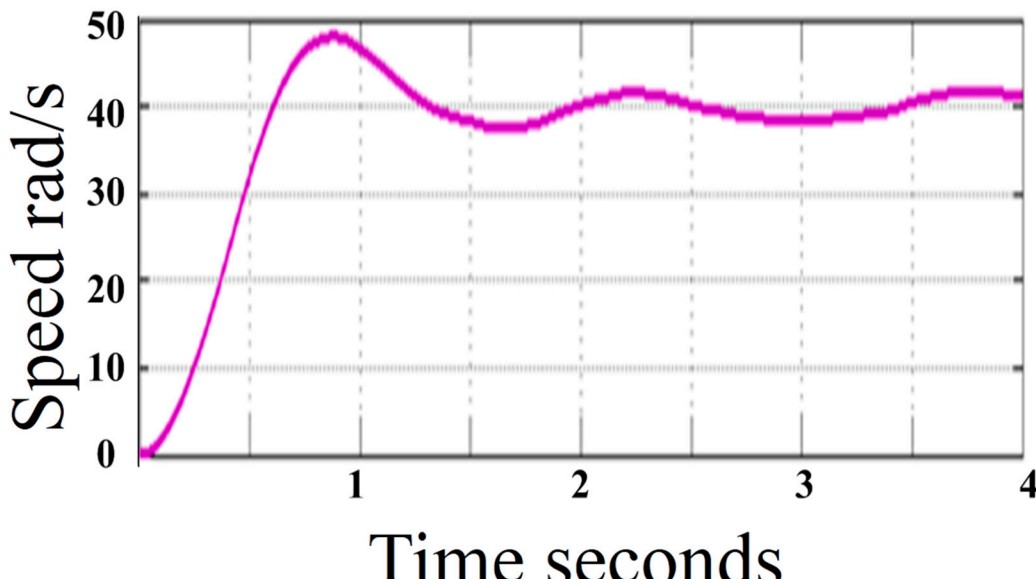

**Figure 15.** Simulation of the best result for ST2FDE with noise 0.9 FLC.

Figure 16 shows the convergence for each of the cases by utilizing the shadowed type-2 fuzzy system. This figure includes the experimentation of the controller without noise, with noise levels of 0.5, 0.7, and 0.9, and the figure clearly shows that when there is more noise the fuzzy system produces better results.

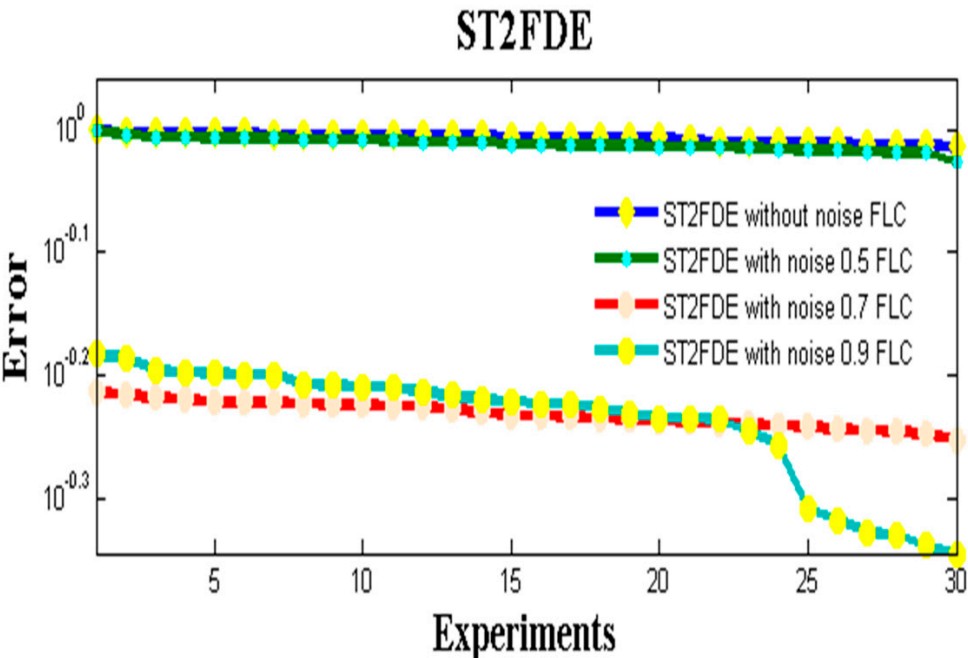

**Figure 16.** Graphical representation of the error using ST2FDE without noise and with different noise levels.

Table 6 shows the results obtained from the experimentation using the general type-2 fuzzy system, without noise in the controller, and with noise levels of 0.5, 0.7, and 0.9, respectively. This table shows the best, the worst, the mean, and the standard deviation results for each case.

**Table 6.** Comparison of results using the GT2FDE optimization of the fuzzy controller.

| Method | GT2FDE without Noise FLC | GT2FDE with Noise 0.5 FLC | GT2FDE with Noise 0.7 FLC | GT2FDE with Noise 0.9 FLC |
|---|---|---|---|---|
| | **GT2FDE** | | | |
| **Best** | $9.73 \times 10^{-01}$ | $9.38 \times 10^{-01}$ | $5.48 \times 10^{-01}$ | $4.35 \times 10^{-02}$ |
| **Worst** | $9.95 \times 10^{-01}$ | $9.91 \times 10^{-01}$ | $6.08 \times 10^{-01}$ | $6.53 \times 10^{-01}$ |
| **Average** | $9.85 \times 10^{-01}$ | $9.75 \times 10^{-01}$ | $5.79 \times 10^{-01}$ | $5.51 \times 10^{-01}$ |
| **Std.** | $5.88 \times 10^{-03}$ | $1.25 \times 10^{-02}$ | $1.70 \times 10^{-02}$ | $7.46 \times 10^{-02}$ |

The visual representation of the best results obtained by experimentation with the fuzzy GT2FDE system is presented in Figures 17–20, which show us the simulation of the controller with the different variants that we have used. In this case, the *x*-axis is the time measured in seconds and the *y*-axis is the speed measured in radians per second, and this is for all the aforementioned figures.

The simulation of the best result obtained in the experimentation is illustrated in Figure 17. The simulation result is $9.73 \times 10^{-01}$ and represents the use of the controller without noise using GT2FDE.

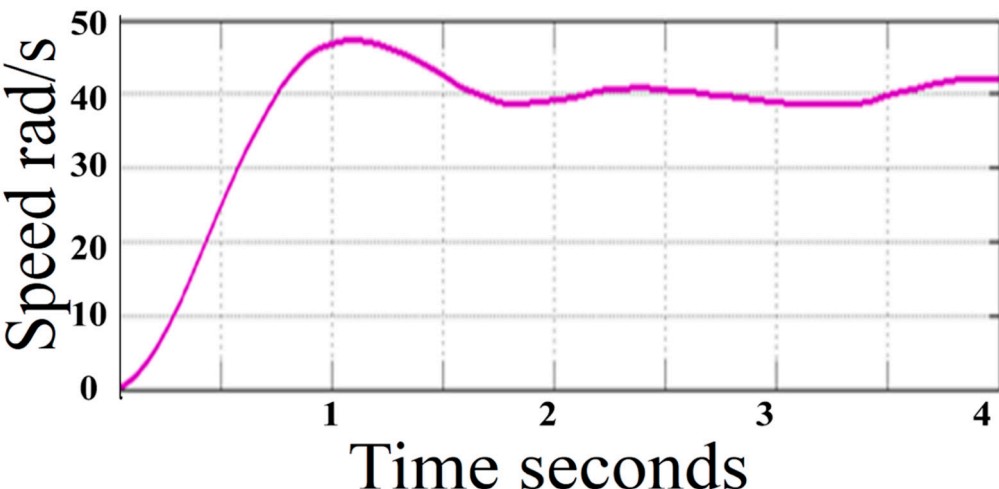

**Figure 17.** Simulation of the best result obtained in the experimentation with GT2FDE without noise in the FLC.

The simulation of the best result obtained in the experimentation is illustrated in Figure 18. The simulation result is $9.38 \times 10^{-01}$ and represents the use of the controller with a noise level 0.5 using GT2FDE.

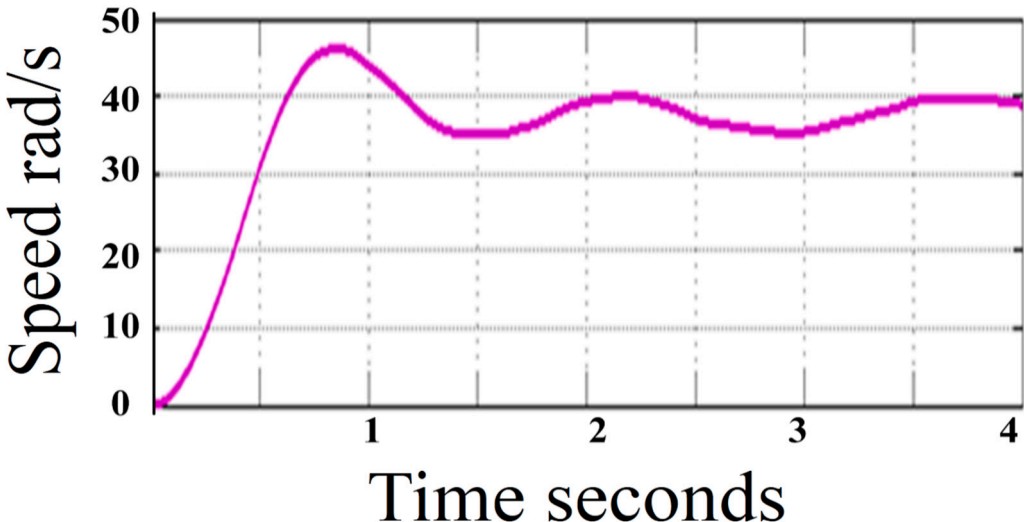

**Figure 18.** Simulation of the best result obtained in the experimentation with GT2FDE with noise of 0.5 in the FLC.

The simulation of the best result obtained in the experimentation is illustrated in Figure 19. The simulation result is $5.48 \times 10^{-01}$ and represents the use of the controller with a noise level 0.7 using GT2FDE.

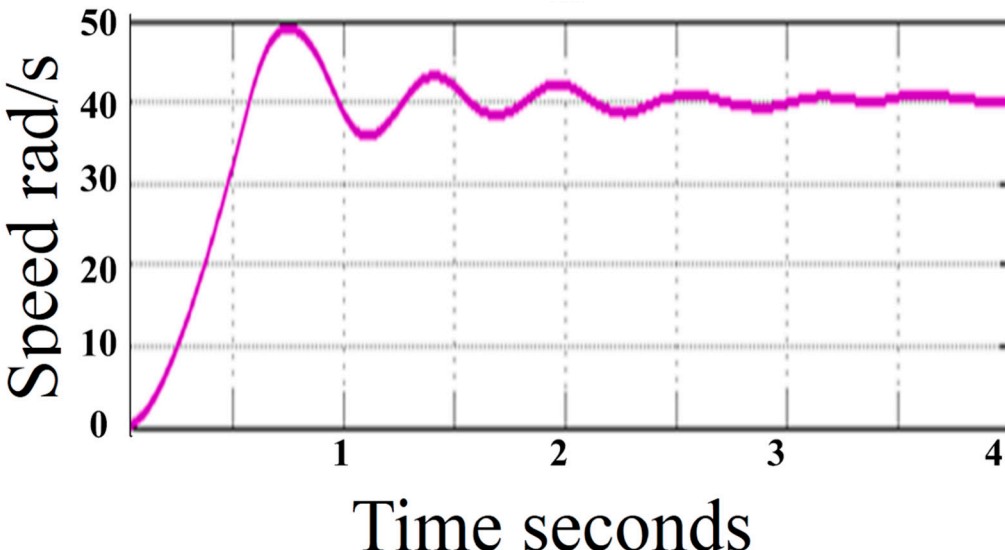

**Figure 19.** Simulation of the best result obtained in the experimentation with GT2FDE with a noise of 0.7 in the FLC.

The simulation of the best result obtained in the experimentation is illustrated in Figure 20. The simulation result is $5.48 \times 10^{-01}$ and represents the use of the controller with a noise level 0.9 using GT2FDE.

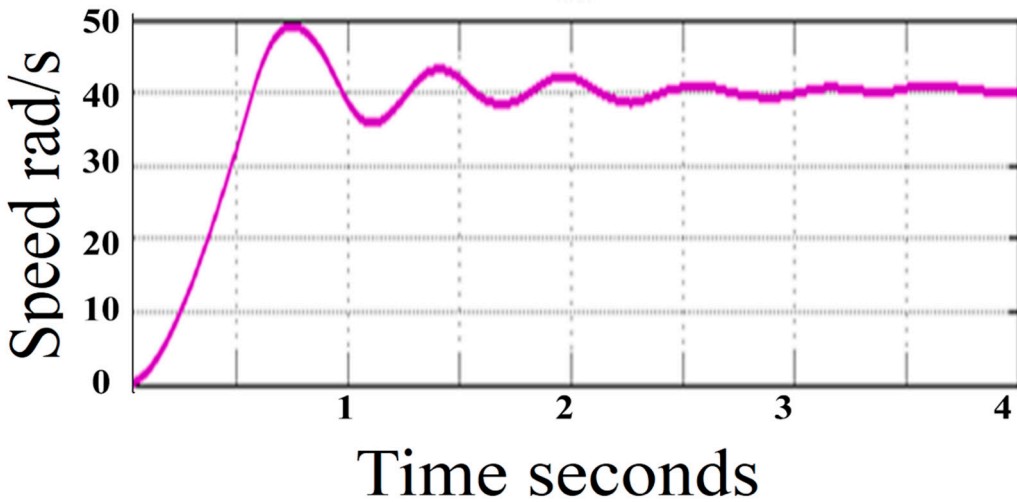

**Figure 20.** Simulation of the best result obtained in the experimentation with GT2FDE with a noise of 0.9 in the FLC.

Figure 21 shows the convergence of each of the cases when using the general type-2 fuzzy alternative. This figure shows the different variants used in the experimentation, and as in the experimentation with the ST2FDE, we can appreciate that with a higher noise level in the controller better results can be obtained.

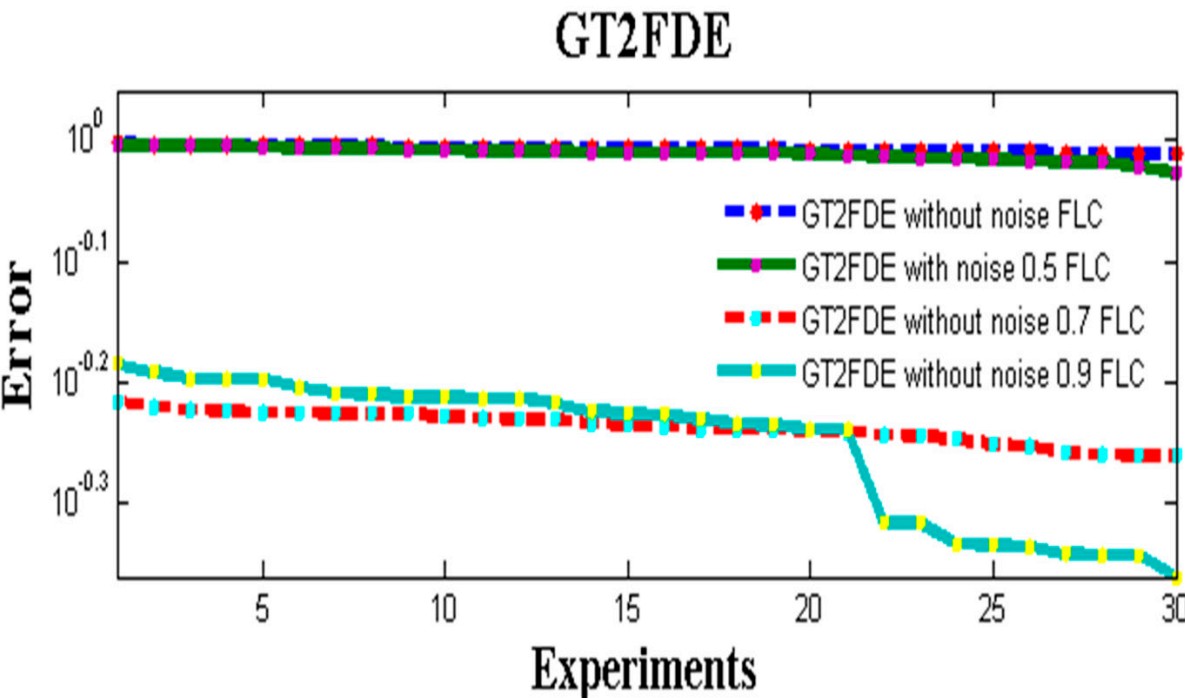

**Figure 21.** Graphical representation of the error using GT2FDE without noise and with different noise levels.

Figure 22 shows a comparison of the best results achieved by each of the variants with noise and without noise using for two fuzzy systems ST2FDE and GT2FDE. We can appreciate that the GT2FDE fuzzy system is slightly better for most of the variants.

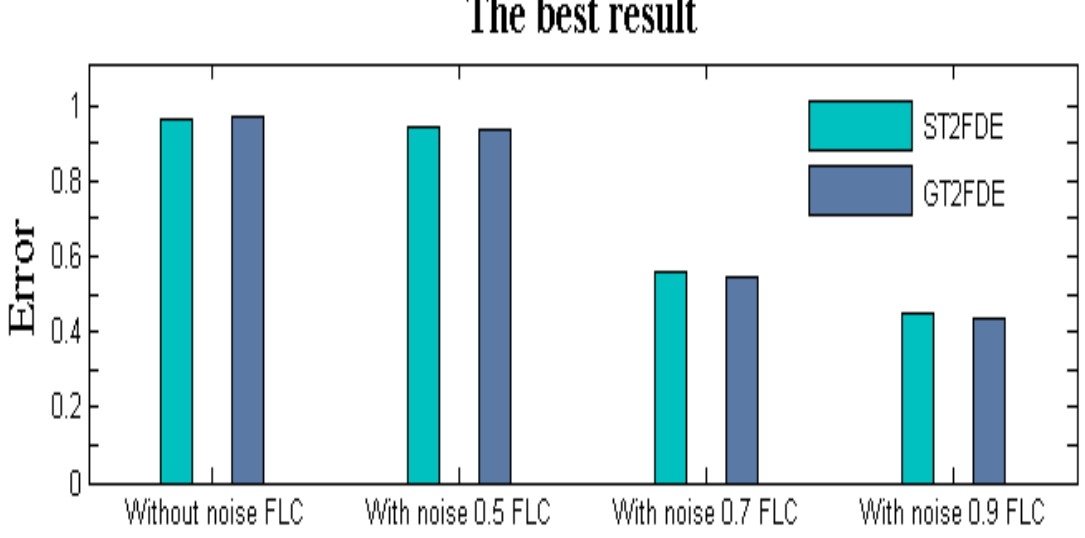

**Figure 22.** Comparison of the best results between ST2FDE and GT2FDE without noise and with different noise levels.

In order to make a decision on which of the used systems has the best result, i.e., which of the two kinds of fuzzy systems is better depending on the achieved error, we carry out a z test.

The parameters used to perform the statistical test are summarized in Table 7.

**Table 7.** Summary of parameters for the z-test.

| Parameter | Value |
|---|---|
| Level of Confidence | 95% |
| Alpha | 0.05% |
| $H_a$ | $\mu_1 < \mu_2$ |
| $H_0$ | $\mu_1 \geq \mu_2$ |
| Critical Value | $-1.645$ |

In this case, $\mu_1$ represents the variants using GT2FDE and $\mu_2$ represents the variants using ST2FDE.

The null and alternative hypotheses that we propose for the statistical test are the following:

Ho: The results of the GT2FDE methodology without noise and with noise are higher than the methodology ST2FDE without noise and with noise.

Ha: The results of the GT2FDE methodology without noise and with noise are lower than the methodology ST2FDE without noise and with noise.

Based on the values shown in Table 7, the rejection zone is for values that are lower than $-1.64$. Equation (24) for calculating the z value of the z-test is presented below:

$$Z = \frac{(\overline{X}_1 - \overline{X}_2) - (\mu_1 - \mu_2)}{\sigma_{\overline{X}_1 - \overline{X}_2}} \tag{24}$$

Table 8 shows the z values obtained for the different statistical tests to compare the performance of the two fuzzy systems.

**Table 8.** Summary of results of the statistical z-tests.

| | Statistical Tests | | | |
|---|---|---|---|---|
| Case Study | $\mu_1$ | $\mu_2$ | Z Value | Evidence |
| | GT2FDE without FCL noise | ST2FDE without FCL noise | 0.5321 | Not Significant |
| Speed control in a D.C. Motor | GT2FDE with FCL 0.5 noise | ST2FDE without FCL 0.5 noise | 0.6398 | Not Significant |
| | GT2FDE with FCL 0.7 noise | ST2FDE without FCL 0.7 noise | $-1.7410$ | Significant |
| | GT2FDE with FCL 0.9 noise | ST2FDE without FCL 0.9 noise | $-1.7018$ | Significant |

Statistical tests show us that when there is a higher noise level, then the general type-2 fuzzy system obtains better results, and the shadowed type-2 fuzzy system is better for lower noise levels.

To verify the efficiency of the GT2FDE fuzzy system, which is statistically better than ST2FDE, we also performed a comparison with the best results obtained in [36]. This previous work used a structure of the fuzzy system that is similar to the one we use here, with one input and one output, where we used the differential evolution algorithm and harmony search (HS).

Table 9 summarizes a comparison of the best results obtained using a high-speed interval Type-2 fuzzy system for parameter adjustment in the DE and HS algorithms [56] and the GT2FDE methodology proposed in this work.

**Table 9.** Comparison between the GT2FDE and other methods.

| | | Method | Best |
|---|---|---|---|
| **D.C. Motor Speed Controller** | **RMSE** | Original DE | $4.72 \times 10^{-01}$ |
| | | DEFIS 1 | $4.57 \times 10^{-01}$ |
| | | DEFIS 2 | $4.80 \times 10^{-01}$ |
| | | DEFIS 3 | $2.36 \times 10^{-01}$ |
| | | Original HS | $4.72 \times 10^{-01}$ |
| | | HSFIS 1 | $4.57 \times 10^{-01}$ |
| | | HSFIS 2 | $4.80 \times 10^{-01}$ |
| | | HSFIS 3 | $2.36 \times 10^{-01}$ |
| | | **GT2FDE with noise 0.9 FLC** | $\mathbf{4.35 \times 10^{-02}}$ |

It is relevant to mention that the comparison is only with the best results since the reference does not provide means and standard deviations to be able to perform a sound statistical test. However, based on the information summarized in Table 9, we can state the proposed method in this paper outperforms the methods presented in [36], which is highlighted in bold.

## 7. Discussion of Results

In this section, the simulation results of the previous section are summarized and discussed with respect to the goals of this article mentioned at the beginning. We first discuss the results of the methods with respect to different levels of noise in the controllers of the motor.

In the case of the simulations carried out with the two kinds of fuzzy systems in DE for a noise level of 0.5, the best results obtained were $9.41 \times 10^{-01}$ and $9.38 \times 10^{-01}$. We can notice that the difference between both is very small, the averages obtained from the experimentation were $9.73 \times 10^{-01}$ and $9.75 \times 10^{-01}$. In the same way, the difference between both averages is minimal, however the statistical test shows that ST2FDE is better than GT2FDE.

The experimentation for the noise level of 0.7 shows the following results $5.59 \times 10^{-01}$ and $5.48 \times 10^{-01}$. However, the averages between both e × periments were $5.86 \times 10^{-01}$ and $5.79 \times 10^{-01}$. Although the difference between the best results is minimal, we can note that, in terms of the averages, there is a significant difference. The statistical test shows that GT2FDE is better than ST2FDE. Finally, in the experimentation with a noise level of 0.9, where the best results were of $4.52 \times 10^{-01}$ and $4.35 \times 10^{-02}$, we can note that there is a difference between these results, for the comparison of the averages obtained, it is $6.13 \times 10^{-02}$ and $7.46 \times 10^{-02}$. It can be observed that there is a difference between both e × periments and finally the statistical test shows that GT2FDE is better than ST2FDE. These results are in accordance with what the literature affirms in most of the works, meaning that general type-2 fuzzy systems are better whenever the noise levels or disturbances are higher, which is what actually occurs in real world problems.

Another way to check the proper functioning of the proposed methodology is the comparison made in Table 9 between high-speed interval type-2 fuzzy system for parameter adjustment in the DE and HS algorithms and the methodology proposed in this work GT2FD. A sample of the best results obtained for comparison are $2.36 \times 10^{-01}$, $2.36 \times 10^{-01}$ and $4.35 \times 10^{-02}$, respectively, where it can be seen that the best result is the one obtained by the methodology proposed in this paper. It is important to mention that the comparison is only with the best result, since the reference does not provide means and standard deviations to perform a statistical test.

## 8. Conclusions

The conclusions for the work presented in this article are summarized below. First of all, we can highlight and affirm that the utilization of general type-2 fuzzy logic is better for higher levels of uncertainty. An analysis of the results between the two kinds of fuzzy systems, ST2FDE and GT2FDE, is as follows. Firstly, the best simulations without noise were of $9.66 \times 10^{-01}$ and $9.73 \times 10^{-01}$, respectively, which are very similar. The same can be observed with the average of the e $\times$ periments where the results were $9.84 \times 10^{-01}$ and $9.85 \times 10^{-01}$, respectively. We can say that the difference between the two is small and the statistical test shows that ST2FDE is better than GT2FDE.

In general, the work carried out shows good results when comparing the two kinds of fuzzy systems and regarding the comparison with the high-speed interval type 2 fuzzy systems combined with the DE and HS algorithms. We can appreciate that we achieved better results with our proposed GT2FDE methodology because the general type-2 fuzzy systems help the differential evolution algorithm a lot in terms of achieving a better performance.

We can affirm that the main contribution of this work is summarized in that we have achieved what has not been done in the previous literature, namely a proposal of the differential evolution combined with shadowed and general type 2 fuzzy systems to dynamically move a parameter of the DE algorithm. The experimentation carried out can help other researchers by providing a guide of the good results obtained when using general type 2 fuzzy systems under high levels of noise.

As future work, we envision that the proposed method could be also applied in other problems in areas such as pattern recognition, time series prediction, and medical diagnosis among others [50–56]. Another important idea is to be able to perform experimentation using the two kinds of fuzzy systems to dynamically adapt the CR (crossover) parameter in some other control problems, to be able to validate with which parameter of the differential evolution algorithm the best results are obtained. In the same way, the experimentation for both parameters to dynamically move (fuzzy system of two outputs) is a task that would be interesting, in order to apply it to control problems, e.g., in robot trajectory tracking.

**Author Contributions:** Contributed to the discussion and analysis of the results in the conclusions, O.C., P.M. and J.S.; contributed to the introduction, simulation results, first; in the implementation of the proposed methodology, second, in finding the optimal design of the type-2 fuzzy systems approach, P.O. All authors have read and agreed to the published version of the manuscript.

**Funding:** This research work did not receive funding.

**Institutional Review Board Statement:** Not applicable.

**Informed Consent Statement:** Not applicable.

**Data Availability Statement:** My manuscript has no associated data.

**Conflicts of Interest:** All the authors in the paper have no conflict of interest.

### Ethical Approval

This article does not contain any studies with human participants or animals performed by any of the authors.

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
