# Peer review of "Differential Evolution with Shadowed and General Type-2 Fuzzy Systems for Dynamic Parameter Adaptation in Optimal Design of Fuzzy Controllers"

_axioms, doi:10.3390/axioms10030194_

Round 1

Reviewer 1 Report

The paper is well written.

Author Response

Response: Thank you for your comments on the paper.

Reviewer 2 Report

-The paper is well written, and it is readable, but there are different parts that must be improved to achieve que expected quality in this journal.

-The methods proposed just can drive the numerical case planted in the paper or it can drive other type of situations.

- I recommend to authors a flowchart in order to explain how the proposal work

- Is necessary a deep literature review in order to present the state of the art. Now just a few intent about it is documented in this work

Author Response

REVIEW 2: -The paper is well written, and it is readable, but there are different parts that must be improved to achieve expected quality in this journal.

Response: Thank you for your comments on the paper.

-The methods proposed just can drive the numerical case planted in the paper or it can drive other type of situations.

Response: Thank you for your comments, the proposed methodology could be applied to different control problems and also different problems of the real world, like the robot trajectory tracking application, actually any problem using fuzzy logic could be adapted to the implementation of the proposed methodology. However, for this work, only the control case was considered in order to be able to measure the performance of the two different types of fuzzy systems.

- I recommend to authors a flowchart in order to explain how the proposal work

Response: Thank you for your comments, a figure has been added with the structure of the proposal for the experimentation carried out in the work.

- Is necessary a deep literature review in order to present the state of the art. Now just a few intent about it is documented in this work.

Response: Thank you for your comments, a literary review was carried out and as a consequence more important references were added to the study that is presented in this work.

Reviewer 3 Report

The work presents a modification of the differential evolution algorithm using shadowed and general type 2 fuzzy systems. The mutation parameter is dynamically moved during the evolution process by using a shadowed and general type-2 fuzzy systems. The main contribution of this work seems to make a performance comparison between using shadowed and general type 2 fuzzy systems as controllers of the mutation parameter in differential evolution. The performance is compared with the single study case problem. The paper seems that has potential however it needs to be improved. The shortcomings must be eliminated before accepting. The list of my comments is as follows:
1. The abstract must be changed. It should smoothly show the background of the work, motivation and justification to undertake this research. The contribution should be more specifically described. The conclusions must be based on results. One study case does not prove anything.  
2. What is the methodological novelty of this work. Is it only a simple comparison of the two methods?
3. The paper is poorly formatted. Please improve it, e.g., Eqs (2), figures and paragraphs.
4. What was the motivation to undertake this research?
5. The figures should be better explained (the captions are also inadequate).
6. Some figures are not needed, e.g., fig 8 etc. and some figures are not readable, e.g. fig9. All figures must be improved (poor quality and sometimes there are very big problems with captions, please see page 13). Fig 20 is very poorly described in the text (not only this one. IT MUST BE IMPROVED.
7. The conclusions are formulated in one study case. The conclusions should be based on results and data. In my opinion, these conclusions are not trustworthy. Therefore, it should be rewritten and interpreted in a good way. The conclusion section should be rewritten. The discussion section is missing.
8. The further research directions should be extended. 
9. Why this research should be publish? Where is new knowledge? It should be shown clearly in the text

Author Response

REVIEW 3: The work presents a modification of the differential evolution algorithm using shadowed and general type 2 fuzzy systems. The mutation parameter is dynamically moved during the evolution process by using a shadowed and general type-2 fuzzy systems. The main contribution of this work seems to make a performance comparison between using shadowed and general type 2 fuzzy systems as controllers of the mutation parameter in differential evolution. The performance is compared with the single study case problem. The paper seems that has potential however it needs to be improved. The shortcomings must be eliminated before accepting. The list of my comments is as follows:

  1. The abstract must be changed. It should smoothly show the background of the work, motivation and justification to undertake this research. The contribution should be more specifically described. The conclusions must be based on results. One study case does not prove anything.  

Response: Thank you for your comments, the abstract was modified describing the background of the work, the motivation and the justification for why this work was carried out and the contribution of the work were described more specifically. The conclusions were modified in such a way as to describe the results and comparisons more clearly.

  1. What is the methodological novelty of this work. Is it only a simple comparison of the two methods?

Response: Thank you for your comments, the novelty of this method is that there are few works in the literature that use shadowed and general type-2 fuzzy systems to dynamically adapt parameters, and the main contribution is to know which of the two kinds of fuzzy systems has better performance and thus be able to be used it in real problems.

  1. The paper is poorly formatted. Please improve it, e.g., Eqs (2), figures and paragraphs.

Response: Thank you for your comments, the equations were modified throughout the document, as well as the figures were modified with better quality

  1. What was the motivation to undertake this research?

Response: Thank you for your comments, the main motivation for doing this work is to be able to determine which of the two types of fuzzy systems is statistically better to be able to be used it in future work in real-world problems.

  1. The figures should be better explained (the captions are also inadequate).

Response: Thank you for your comments, all the figures were modified with a better quality and the captions were modified for a better appreciation of the figures by the readers.

  1. Some figures are not needed, e.g., fig 8 etc. and some figures are not readable, e.g. fig9. All figures must be improved (poor quality and sometimes there are very big problems with captions, please see page 13). Fig 20 is very poorly described in the text (not only this one. IT MUST BE IMPROVED.

Response: Thank you for your comments, an improvement was made to all the figures of the article, the captions of the figures were modified in such a way that they were more legible, also figure 8 was removed as suggested by the reviewer and the descriptions of the figures were modified.

  1. The conclusions are formulated in one study case. The conclusions should be based on results and data. In my opinion, these conclusions are not trustworthy. Therefore, it should be rewritten and interpreted in a good way. The conclusion section should be rewritten. The discussion section is missing.

Response: Thank you for your comments, the conclusions were modified in such a way as to describe the results and comparisons more clearly and also the discussion of results section was added.

  1. The further research directions should be extended. 

Response: Thank you for your comments, the future research directions were extended in the conclusions section.

  1. Why this research should be publish? Where is new knowledge? It should be shown clearly in the text

Response: Thank you for your comments, the findings of the performed study are mentioned in the conclusions in order to be able to show that the performed research has obtained good results and the main contribution of the publication of our work is now clearly mentioned with respect to the existing works in the literature.

Round 2

Reviewer 3 Report

The paper can be accepted after figures improvement. There should be one style of captions and labels. It must be done before final acceptance. I suggest minor acceptance.